# Obesity Prevalence and Associated Factors in Captive Asian Elephants (*Elephas maximus*) in China: A Body Condition Assessment Study

**DOI:** 10.3390/ani14243571

**Published:** 2024-12-11

**Authors:** Yinpu Tang, Ting Jia, Fangyi Zhou, Liang Wang, Ziluo Chen, Li Zhang

**Affiliations:** 1Key Laboratory for Biodiversity Science and Ecological Engineering, Ministry of Education, College of Life Sciences, Beijing Normal University, Beijing 100875, China; yinpu_tang@mail.bnu.edu.cn (Y.T.); 202227200021@mail.bnu.edu.cn (L.W.); 202421200065@mail.bnu.edu.cn (Z.C.); 2Beijing Key Laboratory of Captive Wildlife Technologies, Beijing Zoo, Beijing 100044, China; jiating_2005@163.com; 3Xishuangbanna Wild Elephant Valley Co., Ltd., Jinghong 666100, China; zhoufangyi@mail.bnu.edu.cn

**Keywords:** Asian elephant (*Elephas maximus*), captive management, obesity, body condition assessment, China

## Abstract

Asian elephants are one of the most widely recognized species in zoos. However, due to the tremendous differences between the captive environment and their native habitats, captive elephants usually encounter physical health challenges such as obesity. Obesity is a common but often overlooked problem in captive elephants, which leads to being overweight and threatens the elephants' athletic and reproductive health. In cases where direct measurements of body mass are impractical, visual assessments of body condition serve as an effective indirect method for evaluating obesity in these animals. In this study, we conducted a comprehensive field survey of captive Asian elephants in China, aiming to identify the contributive factors to their obesity through visual evaluations and statistical analyses. Our results suggest that inadequate outdoor time significantly restricts exercise opportunities for captive elephants, which plays a crucial role in the development of obesity together with a calorie-dense diet and limited outdoor space. These findings support further exploration of solutions to overweight in captive elephants that incorporate enrichments, diet, veterinary care, and positive reinforcement training.

## 1. Introduction

Asian elephants (*Elephas maximus*), the largest terrestrial animals in Eurasia, are classified as Endangered (EN) on the IUCN Red List and listed under CITES Appendix I. In the wild, Asian elephants inhabit 13 countries across South and Southeast Asia, with an estimated population of 48,323 to 51,680 individuals [1]. Approximately 16,000 Asian elephants are kept in captivity (e.g., in zoos and wildlife sanctuaries) or semi-captivity (e.g., within timber enterprises or by private owners), where they are primarily used for transport or as draught animals; only about 8.75% (<1400) reside in zoos [2,3,4].

The research shows that captive elephants are susceptible to numerous health issues, leading to reduced lifespans. Combined with low reproductive rates, zoo elephant populations are generally not self-sustaining [2,5,6,7]. As a result, the reproductive health of captive Asian elephants and its influencing factors have been focal points of management and care strategies [8,9,10,11].

However, in comparison to physical injuries or infectious diseases, obesity—despite its prevalence in captive wildlife—has received limited attention from caregivers [12,13]. Overweight due to obesity will cause abnormal stress on the leg and foot joints, leading to arthritis and even disability in captive elephants [14]. Moreover, obesity will also increase the risks of dystocia and stillbirth for pregnant elephants in captivity [15].

From an energy budget perspective, obesity can be understood as an imbalance between nutritional intake and physical activity [16,17]. In captive Asian elephants, an improper diet and lack of exercise often contribute to this imbalance [18]. A survey by the Association of Zoos and Aquariums (AZA) in the USA found that among 108 captive Asian elephants in 65 North American zoos, 75% of females and 65% of males were overweight or obese to varying degrees [19]. The research on captive Asian elephants in European and American zoos, as well as those used in Thai tourism, suggests that obesity can be indicative of metabolic disorders, which are closely linked to reduced fertility in both males and females [20,21,22]. In contrast, studies on free-ranging Asian elephants in Sri Lanka show that a healthy body condition and sufficient physical activity are positively associated with female reproductive success and that relative body weight (kg/cm at shoulder) correlates with reproductive output [23].

China’s history of raising Asian elephants in zoos dates back to 1907 [24], with the first case of artificial breeding occurring in 1966 [25]. Since the 1980s, China has imported a large number of Asian elephants, and their population has grown with successful breeding efforts in various facilities. Despite these advancements, the management of captive Asian elephants in China lags in areas such as research and welfare, particularly concerning the impact of obesity. This study aims to investigate the factors influencing obesity in captive Asian elephants in China. It further explores the challenges facing these elephants in captivity and offers practical recommendations to improve their welfare and physical condition.

## 2. Materials and Methods

### 2.1. Ethics Statement

All data collected in this study were solely for scientific purposes. The Endangered Species Scientific Commission of P.R. China (ESSC, P.R.C.) and the Chinese Association of Zoological Gardens (CAZG) reviewed and approved all procedures and issued permits for this study in the specified elephant-raising facilities. As the study employed observation-based methods without direct contact with animals or use of tissue samples, approval from an Institutional Animal Care and Use Committee or equivalent animal ethics committee was not required.

### 2.2. Animals

From January 2017 to April 2019, we recorded data for a total of 204 captive Asian elephants (♂:♀ = 88:116), representing about 64% of the captive Asian elephant population in China. Data collection involved on-site investigations across 43 facilities, including 42 zoos across the country and China Asian Elephant Conservation and Rescue Center at the Wild Elephant Valley in Xishuangbanna. Through interviews with facility staff (veterinarians, curators, and caregivers) and the examination of animal records, we documented the sex and age of each elephant. However, the ages of 31 individuals (♂:♀ = 13:18) were unknown (see Figure 1 for details).

### 2.3. Study Areas

#### 2.3.1. Zoos

In this study, a total of 42 zoos across 25 provinces, municipalities, and autonomous regions were visited. Each zoo typically opens daily from 9:00 a.m. to 5:00 p.m. During warm weather, elephants are displayed in outdoor enclosures during opening hours and are brought indoors once the zoo closes. Due to a lack of outdoor heating equipment, most zoos—especially those in Northern China—restrict outdoor elephant exhibits when temperatures fall (e.g., ≤15 °C), resuming outdoor displays only as temperatures rise again the following year. Each elephant receives a specified amount of feed daily, divided into two feeding intervals in the morning and afternoon.

#### 2.3.2. Wild Elephant Valley

The Wild Elephant Valley (100°51′33.2172″ E, 22°10′39.3636″ N) is situated in the Xishuangbanna National Nature Reserve, Yunnan Province. Serving as both a tourist attraction and an elephant rescue center, it houses a diverse group of elephants, including those imported and trained for performance, as well as captive-born and rescued elephants. Outside of performance time (about two hours daily), the elephants are allowed to roam and forage in a nearby natural forested area of 340 mu (approximately 226,667 m^2^), which includes 8000 m^2^ of surface water and 2800 m^2^ that intersect with local villages. Elephants here are provided with four tons of fresh grasses daily, primarily elephant grass (*Cenchrus purpureus*), occasionally supplemented with ryegrass (*Leymus chinensis*) and corn stalks, along with 100 kg of seasonal fruit. During feeding times, elephants tend to form small groups of four to ten individuals, with each group sharing its own pile of feed, separate from other groups.

### 2.4. Obesity Status Assessment: Body Condition Scoring

Due to limited training and equipment in most study areas, we were unable to directly weigh the elephants. Consequently, we used a visual body condition assessment—a widely used method for evaluating overall animal health—to represent the obesity status of the elephants. Initially developed for assessing large livestock, such as cows and horses, this method has been adapted for use with wildlife in both natural habitats and captivity [27,28,29,30]. In our study, we determined each elephant’s body condition score (BCS) following the criteria established by Fernando et al. (2009) (Table 1), and hypothesized that a higher BCS would indicate a higher obesity status [31,32].

Photographs and videos of each elephant were captured from multiple angles at each facility. Selected images for body condition assessment were taken by the first, third, and fourth authors, including photographs and video screenshots, which met the following criteria: (i) clearly identifiable individuals; (ii) elephants in a standing or moderate walking position to enable a reliable assessment; (iii) sufficient visibility of prominent bone structures (skull, shoulder girdles, vertebral column, ribs, pelvic girdles, and backbone); and (iv) adequate resolution to discern typical skin wrinkles. Images were excluded if (v) distinct shade patterns or materials, such as hay or straw, obscured the elephant’s back, or if lateral sunlight diminished the image contrast, potentially affecting the evaluation accuracy. All images were scored by the first author. Individuals were identified based on morphological features, such as facial appearances, ear shape, tusks, tail hair, and scars or injuries on the skin. In the body condition scoring of every individual, images were presented randomly. Each elephant was evaluated from both sides—left and right—and the BCS was calculated as the rounded average of the scores from the left and right images. Elephants with a BCS of 7 or higher were classified as obese.

### 2.5. Potential Related Factors of BCS

Including sex and age, we investigated seven potential related factors of BCS (Table 2) in total, five of which were associated with diet and exercise.

#### 2.5.1. Potential Related Factors of BCS Associated with Diet

There were two potential related factors of BCS associated with diet: daily feed supply (kg) and the proportion of high-calorie feed (%).

Our investigation showed that for all 43 facilities, the feed of elephants was mainly divided into three categories: (i) herb forage, including various grasses and hays (e.g., Chinese rye grass, Sudan grass, medick, straw, etc.); (ii) juicy fodder (seasonal fruits and vegetables); and (iii) pellet feed, the main ingredients of which were corn kernels, wheat bran, and soybean meal. Here, we define ‘high-calorie feed’ as the sum of juicy fodder and pellet feed. For elephants in zoos, we collected the weight of three categories of feed above daily supplied for each elephant by interviewing staff members and consulting feeding documentation, then calculated the daily feed supply and the proportion of high-calorie feed of each elephant (Table 2A). For elephants in the Wild Elephant Valley, since it was challenging to ensure the exact daily feed supply for each elephant, we hypothesized that its daily total feed supply was distributed equally to each elephant (Table 2B).

Here, we hypothesized that both a larger daily feed supply and a higher proportion of high-calorie feed could associate with higher BCS in an elephant.

#### 2.5.2. Potential Related Factors of BCS Associated with Exercise

Three factors potentially associated with body condition score (BCS) and exercise were identified: outdoor enclosure area (m^2^ per elephant), outdoor time (months per year), and foot disorders. To minimize stress on the elephants, we refrained from direct contact during our investigation, making it challenging to obtain precise exercise data through activity-tracking bracelets or by measuring oxygen consumption with wearable air-collecting devices [33,34]. However, we hypothesized that larger outdoor enclosures and extended outdoor time, compared to confined indoor spaces, would offer elephants greater room for movement and the expression of natural behaviors, thereby increasing exercise levels and energy expenditure. Additionally, we posited that foot disorders could impair mobility, reducing exercise and energy expenditure.

We examined the outdoor enclosures available for elephants at each facility (Table 2). The outdoor enclosure area was measured using Google Earth Pro 7.3.2.5766 by selecting the relevant region on the map. For the Wild Elephant Valley, local villages and surface water bodies were excluded from the enclosure area calculation. Outdoor time for zoo-housed elephants was determined through interviews with staff and by consulting monthly exhibit records.

For each elephant, we took close-up photos of all four feet, recording any visible foot disorders, which included: (i) overgrown nails, (ii) nail cracks, (iii) overgrown cuticles, and (iv) joint deformities (Figure 2, Table 2). If none of these conditions were observed, we classified the elephant’s feet as healthy (Figure 2A).

Our hypothesis suggested that a larger outdoor enclosure area and longer outdoor time would correlate with a lower BCS in elephants, whereas the presence of foot disorders might correlate with a higher BCS.

### 2.6. Statistical Analysis

All statistical analyses were performed using R version 4.2.2. Sex was included as a categorical variable in all calculations, with males and females encoded as 1 and −1, respectively. Individuals with unknown ages were excluded from the dataset.

To investigate the relationship between the body condition score (BCS) and its potential influencing factors, we conducted a principal component analysis (PCA) using the ‘prcomp’ package. This analysis included seven independent variables—sex, age, daily feed supply, proportion of high-calorie feed, outdoor enclosure area, outdoor time, and foot disorder—along with BCS. The component scores and loadings were visualized with a biplot.

For further analysis, we used the ‘lmer’ package to construct a stepwise multivariable linear mixed model (LMM) with eight variables (sex, age, facility category, daily feed supply, proportion of high-calorie feed, outdoor enclosure area, outdoor time, and foot disorder) as fixed effects and different facilities as random effects. The Akaike Information Criterion (AIC) was applied to determine the optimal model. We then examined interaction effects between variables in the optimal model to establish the best interaction model. Finally, we assessed the significance of the relationships between BCS and its potential influencing factors through significance testing.

## 3. Results

### 3.1. Overview of Captive Asian Elephants’ Body Condition in China

We evaluated the body condition scores (BCSs) of 204 captive Asian elephants across 43 facilities in China (Table 3), finding that 72.55% of these elephants were classified as obese to varying degrees (BCS ≥ 7). On average, elephants housed in zoos had a significantly higher BCS compared to those in the Wild Elephant Valley (Independent Sample T-Test, *p* < 0.001).

### 3.2. Relationship Between BCS and Potential Influencing Factors

Principal component analysis (PCA) reduced the dataset from eight variables (seven independent variables and BCS) to two principal components. The first principal component (PC1) and the second principal component (PC2) accounted for 38.73% and 16.76% of the variance, respectively, with the biplots of PC1 and PC2 representing over half of the dataset’s variance (Figure 3). This analysis suggested a positive correlation between BCS and the proportion of high-calorie feed. In contrast, both outdoor enclosure area and outdoor time were strongly negatively correlated with BCSs.

For individuals with a known age (n = 173), the optimal linear model selected through stepwise regression included three BCS predictor variables: sex (*p* < 0.01), outdoor enclosure area (*p* < 0.001), and outdoor time (*p* < 0.001) (Appendix A). ANOVA tests revealed that models including the interaction of sex with outdoor enclosure area and outdoor time, or the interaction of sex with outdoor time and outdoor enclosure area as independent variables, were statistically equivalent (Appendix A). Considering that facilities typically allow both sexes of elephants to be outdoors in suitable weather, we developed a mixed regression model with the BCS as the dependent variable and four predictor variables: (i) sex; (ii) outdoor enclosure area; (iii) outdoor time; and (iv) the interaction between sex and outdoor enclosure area (“sex × outdoor enclosure area”) as fixed effects, with facility as a random effect. The results indicate significant negative correlations between the BCS and outdoor enclosure area, outdoor time, and the interaction between sex and outdoor enclosure area (*p* < 0.05) (Table 4), suggesting that a larger outdoor area and longer outdoor time are associated with a lower BCS.

Considering the notable differences in outdoor enclosure areas between the Wild Elephant Valley and zoo-housed elephants, we applied the same model selection approach to the zoo population with known ages (n = 144) (Appendix A). The results indicate that only outdoor time is negatively correlated with the BCS (*p* < 0.01) (Table 5).

## 4. Discussion

This study examined the relationship between potential influencing factors and obesity status in captive Asian elephants in China through body condition assessment and linear regression. Results indicate that increased outdoor time is associated with a lower BCS. The following discussion addresses various factors related to diet, exercise, and facility conditions affecting these elephants.

### 4.1. Enclosures, Exhibition, and Obesity of Captive Elephants in Zoos

Our linear regression analysis showed that, among elephants with known ages, both outdoor enclosure area and outdoor time were significantly negatively correlated with BCS. However, for zoo-housed elephants specifically, no significant correlation was observed between outdoor enclosure area and BCS. A likely explanation for this discrepancy is that the outdoor enclosures provided by Chinese zoos are generally too small.

To promote welfare, facilities should aim to provide the largest possible outdoor enclosures and incorporate enrichment elements that encourage natural behaviors and physical activity, which may help reduce or prevent obesity [35]. A study on captive Asian elephants in European zoos found that females in smaller enclosures were more susceptible to obesity [36]. Although the Chinese Association of Zoological Gardens (CAZG) has not set strict requirements for the outdoor space provided for captive elephants, it does recommend a minimum outdoor enclosure area of 170 m^2^ per elephant [37]. Despite these recommendations, our findings reveal that nearly half (48.82%) of elephants in zoos do not meet CAZG’s recommended standard. Given the challenge most zoos face in expanding outdoor spaces in the short term, we suggest that Chinese zoos prioritize maximizing the use of limited outdoor enclosures through diverse enrichment options, such as pools, sand pits, wallows, scratching posts, shade canopies, and foraging devices. Additionally, maintaining a suitable population size within enclosures is essential, as overcrowding (e.g., more than ten elephants in one enclosure) limits available space per elephant, while housing only one elephant can negatively impact its mental health over time due to the species’ complex social needs.

Numerous studies highlight that inactivity and prolonged indoor confinement increase the risk of degenerative musculoskeletal conditions in captive elephants, exacerbating obesity and potentially leading to disability or euthanasia [38,39,40]. Regarding the relationship between outdoor time and obesity, the research on captive Asian elephants in North American zoos found that staff-directed walking exercises totaling 14 or more hours per week significantly reduce obesity risk [19]. Conversely, a 10-month observational study on captive Asian elephants in a British zoo showed that, during the average 6.5 h of daily outdoor time, adult elephants spent most of it feeding, standing still, or resting, with only 6–19% of the time spent walking [41].

We recommend that zoo staff consider measures to increase the outdoor time of captive elephants, such as allowing elephants to freely enter and exit outdoor enclosures after zoo hours. Facilities might also install outdoor heat sources to extend outdoor time during colder temperatures.

### 4.2. Diet and Obesity of Captive Elephants

While our statistical analysis indicated that daily feed supply and the proportion of high-calorie feed were not significantly related to BCS, we were unable to measure the exact feed intake of elephants in this study. Further research is needed to clarify the effects of diet on obesity in captive elephants in China. It is recommended that cellulose content in elephant diets should be at least 30%, with herb forage comprising no less than 70% of the total daily feed mass to prevent digestive issues [42,43]. High-calorie feed items (e.g., fruits, vegetables, and pellets) should be carefully regulated, with essential trace elements supplemented as necessary to prevent obesity and avoid digestive issues caused by excessive protein [44]. Our investigation revealed that over a quarter (27.45%) of elephants had diets with more than 30% high-calorie feed. Additionally, some zoos add starchy foods, such as pumpkins and sweet potatoes, to elephants’ diets during winter months. Given that elephants remain indoors with limited exercise during this time, increased nutrient intake may contribute to a higher BCS. We suggest that zoos control high-calorie feed supplies, use feeding devices for enrichment, and implement varied feeding schedules to aid in managing the body weight of captive elephants [45,46].

### 4.3. Foot Health of Captive Asian Elephants in China

Although statistical analysis showed that foot disorders were not significantly related to the BCS, our investigation found that 58.24% of zoo elephants had foot disorders of varying severity. Studies in European zoos indicate that time spent indoors and on hard surfaces is positively correlated with foot disorders, while natural, soft substrates (e.g., sand) can mitigate these issues [47]. However, all 42 zoos we investigated used cement flooring in indoor spaces. European and North American zoos often provide regular foot health checks supported by protected contact and positive reinforcement training [48]. In contrast, less than 25% of Chinese zoos had the necessary equipment, such as training walls, or trained personnel for regular foot care. Many foot issues require diagnosis by CT scans or by observing posture during movement, rather than relying solely on visual inspection [49,50]. This suggests that foot health issues among captive elephants in China may be more serious than currently observed. We recommend that facilities increase awareness of the importance of foot health and promote positive reinforcement training for routine foot examinations.

### 4.4. Elephants in the Wild Elephant Valley

Our investigation and statistical analysis showed that elephants in the Wild Elephant Valley had significantly lower BCSs on average and no visible foot disorders, likely due to environmental and husbandry conditions that resemble natural Asian elephant habitats. However, it is difficult to duplicate similar environmental conditions in the Wild Elephant Valley for most zoos in China. Even so, the feed supply in Wild Elephant Valley may be worth referring to for elephant caregivers in zoos.

### 4.5. Animal Welfare Implications

This study represents the first investigation into obesity and foot health among captive Asian elephants in China. However, limitations remain. For example, not all captive Asian elephants in China were included, and we were unable to obtain tissue samples for biochemical analyses due to limited veterinary facilities. Although body condition scoring is less precise than direct weighing and serum leptin measurements for assessing obesity [33], we hope our study draws attention to the welfare concerns, including obesity, of captive Asian elephants in China. We also aim for our findings to inspire improved husbandry practices for captive elephants and to serve as a reference for future welfare studies on captive megaherbivores.

## 5. Conclusions

Our study suggests that limited outdoor time may be a primary contributing factor to the high-body-condition scores observed among captive Asian elephants in China. We recommend that facilities housing elephants implement measures to increase outdoor time. Severely overweight elephants should be allowed more than an hour of outdoor exercise every day under the supervision of keepers if the weather allows this, rather than just wandering around the exhibit. Additionally, a balanced diet with feeding enrichment, combined with regular foot health checks, may further support effective body weight management in captive elephants.

## Figures and Tables

**Figure 1 animals-14-03571-f001:**
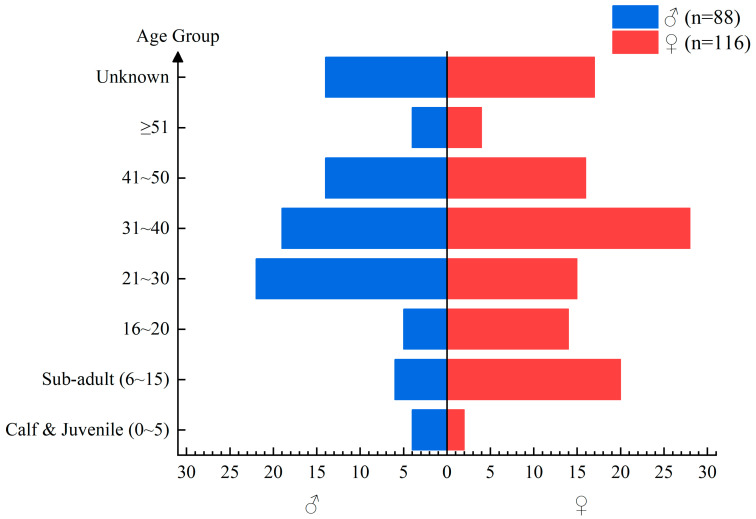
Age structure of 224 captive Asian elephants recorded from zoos and Wild Elephant Valley in China from January 2017 to April 2019. Classification of age groups is referred to in the criteria in Arivazhagan and Sukumar (2008) [26].

**Figure 2 animals-14-03571-f002:**
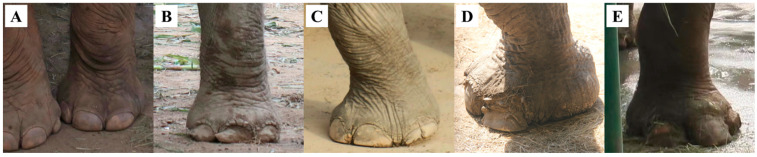
Samples of visible foot disorders of captive Asian elephants (Elephas maximus) recorded during on-site investigations: (**A**) feet with healthy appearance; (**B**) overgrown nails; (**C**) nail cracks; (**D**) overgrown cuticles; (**E**) joint deformation.

**Figure 3 animals-14-03571-f003:**
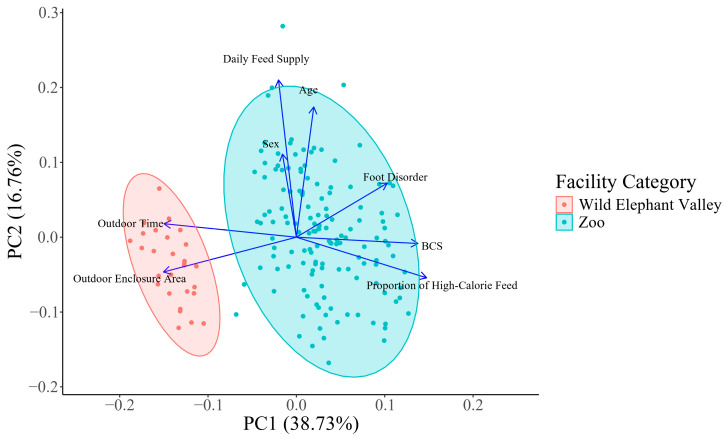
Biplot of principal component analysis (PCA) of BCS and its potential relative factors for all individuals with a known age (n = 173). Points in the plot represent individual elephants, and eigenvectors show relations between the variables. Eigenvectors pointing in similar directions are positively correlated in the first two principal components, whereas eigenvectors pointing in opposite directions are negatively correlated. The length of each eigenvector represents its contribution to the corresponding variable.

**Table 1 animals-14-03571-t001:** Criteria of visual body condition scoring (BCS) used in this study to evaluate Asian elephants (*Elephas maximus*) for obesity (from Fernando et al. 2009) [31].

Score	Characters
1	All ribs (shoulder to pelvis) visible, some ribs prominent (spaces in between sunken in)
3	Some ribs visible (spaces in between not sunken in), shoulder and pelvic girdles prominent
5	Ribs not visible, shoulder and pelvic girdles visible
7	Backbone visible as a ridge, shoulder and pelvic girdles not visible
9	Back rounded, thick rolls of fat under neck

When the body condition of an elephant is intermediate between two criteria, an intermediate point score (i.e., 2, 4, 6, and 8 points) should be assigned. To accommodate the possibility of greater variation, the range of scores extend from 0 to 10.

**Table 2 animals-14-03571-t002:** Overview of seven potential relative factors of BCS for elephants in different facilities: (**A**) zoos; (**B**) Wild Elephant Valley.

(**A**)						
**Sex**	**Age** **(year, Mean ± SE)**	**Daily Feed Supply** **(kg, Mean ± SE)**	**Proportion of High-Calorie Feed** **(%, Mean ± SE)**	**Outdoor Enclosure Area** **(m^2^ per elephant, Mean ± SE)**	**Outdoor Time** **(months per year, Mean ± SE)**	**Proportion of Foot Disorder**
Both	29.27 ± 0.092 (n = 144)	136.56 ± 0.302 (n = 170)	24.05 ± 0.065 (n = 170)	307.38 ± 2.038 (n = 170)	8.19 ± 0.011 (n = 170)	58.24% (99/170)
Male	30.31 ± 0.218 (n = 62)	144.15 ± 0.784 (n = 73)	23.79 ± 0.152 (n = 73)	388.98 ± 5.919 (n = 73)	8.18 ± 0.025 (n = 73)	50.68% (37/73)
Female	28.49 ± 0.161 (n = 82)	130.84 ± 0.472 (n = 97)	24.24 ± 0.113 (n = 97)	245.96 ± 2.576 (n = 97)	8.20 ± 0.021 (n = 97)	63.92% (62/97)
(**B**)						
**Sex**	**Age** **(year, Mean ± SE)**	**Daily Feed Supply** **(kg, on average)**	**Proportion of High-Calorie Feed** **(%, on average)**	**Outdoor Enclosure Area** **(m^2^ per elephant, on average)**	**Outdoor Time** **(months per year)**	**Proportion of Foot Disorder**
Both	25.86 ± 0.417 (n = 29)	120.56 (n = 34)	2.44 (n = 34)	6349.02 (n = 34)	12 (n = 34)	0.00% (0/34)
Male	29.67 ± 0.938 (n = 12)					
Female	23.18 ± 0.720 (n = 17)					

**Table 3 animals-14-03571-t003:** Overview of captive Asian elephants’ body condition score (BCS) in China.

Facility	Sex	BCS Mean ± SE	BCS Median	Proportion of BCS ≥ 7
All	Both (n = 204)	7.07 ± 0.007	7	72.55%
	Male (n = 88)	6.84 ± 0.017	7	69.32%
	Female (n = 116)	7.24 ± 0.010	7	75.00%
Zoo	Both (n = 170)	7.43 ± 0.006	7	82.35%
	Male (n = 73)	7.33 ± 0.013	7	80.82%
	Female (n = 97)	7.51 ± 0.010	7	83.51%
Wild Elephant Valley	Both (n = 34)	5.26 ± 0.044	5.5	23.53%
	Male (n = 15)	4.47 ± 0.100	4	13.33%
	Female (n = 19)	5.89 ± 0.061	6	31.58%

**Table 4 animals-14-03571-t004:** Significance test of BCS predictor variables for all individuals of a known age (n = 173).

Fixed Effects	Estimate	SE	df	t-Value	*p*-Value (>|t|)
(Intercept)	9.373 × 10	3.697 × 10^−1^	1.139 × 10^2^	25.350	<2 × 10^−16^ ***
Sex	−1.034 × 10^−1^	1.822 × 10^−1^	4.780 × 10^1^	−0.567	0.57311
Outdoor Enclosure Area	−1.378 × 10^−4^	4.868 × 10^−5^	1.572 × 10^2^	−2.831	0.00524 **
Outdoor Time	−2.133 × 10^−1^	4.296 × 10^−2^	9.380 × 10^1^	−4.964	3.08 × 10^−6^ ***
‘Sex’ × ‘Outdoor Enclosure Area’	−2.540 × 10^−4^	8.103 × 10^−5^	4.838 × 10	−3.134	0.02702 *

Significance code: ‘***’: *p* < 0.001; ‘**’: *p* < 0.01; ‘*’: *p* < 0.05.

**Table 5 animals-14-03571-t005:** Significance test of BCS predictor variables of BCSs for individuals with a known age in zoos (n = 144).

Fixed Effects	Estimate	SE	df	t-Value	*p*-Value (>|t|)
(Intercept)	9.01193	0.45191	24.58927	19.942	<2 × 10^−16^ ***
Outdoor Time	−0.18904	0.05397	22.61886	−3.503	0.00195 **

Significance code: ‘***’: *p* < 0.001; ‘**’: *p* < 0.01.

## Data Availability

The authors confirm that the data supporting the findings of this study are available within the article and its Appendix A.

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
