# Peer review of "Obesity Prevalence and Associated Factors in Captive Asian Elephants (Elephas maximus) in China: A Body Condition Assessment Study"

_animals, 2024, doi:10.3390/ani14243571_

Round 1
Reviewer 1 Report
Comments and Suggestions for Authors
This is the first publication of BCS of Asian elephants in human care in China. I found some minor issues with this manuscript. While it is well written, I have some questions/suggestions, which could raise the level of confidence in the manuscript as it is currently presented.
1. One issue is that while a citation is used that, described BCS in a small free ranging population in China, no comparison of the wild elephants BCS with the semi-captive population of Wild Elephant Valley is brought forward in the discussion.
2. Lines 51 and 267: a citation by Morfeld et. al. (2016) is not listed in the references.
3. Line 98: I suggest changing from “meals served” to feeding intervals….
4. Line 124: I suggest adding an asterisk in the table to which refers to the current asterisk.
5. Who obtained the Photographs and videos? ‘We’ was used but later only the first author provided a BCS for the manuscript. If the person or persons who obtained the photographs was the first author, it would be an improvement to the credibility to have more than one person score the elephants and combine scores. In addition, why were videos obtained? There is no mention of using the videos in the manuscript.
6. In the same paragraph each elephant was scored twice (once on each side} by the same person. Adding a description of how the photographs were presented for scoring would help in clarifying the method of determining BCS. Were the photographs presented randomly for scoring without identification, and later decoded for analysis. A question remains, were the images of the same elephant sequentially presented?
7. Lines 312-313: There was no data presented which would support the statement in this sentence. I highly request removing this sentence.
8. Lines 313-317: The statement regarding passing EEHV from Wild Elephant Valley is misstating the information found in the citation of Yang et. al. 2022. EEHV has been found in all Asian elephant populations studied in range countries and was found in the small sample size of wild Asian elephants in China reported by Yang et. al. I highly request removing the sentence from 313-317.
By making these changes, I believe this is a manuscript which will adding to the information available on captive Asian elephants in China.
Author Response
Comments from Reviewer 1:
This is the first publication of BCS of Asian elephants in human care in China. I found some minor issues with this manuscript. While it is well written, I have some questions/suggestions, which could raise the level of confidence in the manuscript as it is currently presented.
- One issue is that while a citation is used that, described BCS in a small free ranging population in China, no comparison of the wild elephants BCS with the semi-captive population of Wild Elephant Valley is brought forward in the discussion.
Response: Thanks for your comment.
The objects of our study are captive Asian elephants in China. We did not cite BCS-related studies on free-ranging elephants in China. In Line 55, we cited a former study on free-ranging Asian elephants in Sri Lanka for reference.
Considering the differences in multiple dimensions (such as habitat use, food availability, social structure, etc.) between free-ranging and captive elephants, although the environment of Wild Elephant Valley is similar to the natural habitat of free-ranging Asian elephants, we did not discuss the wild elephants BCS with the semi-captive population of Wild Elephant Valley in this manuscript. Moreover, there is currently a lack of large-scale studies on BCS or the obesity status of free-ranging Asian elephants in China.
Therefore, we appreciate your suggestion which provides a meaningful direction for our future research.
- Lines 51 and 267: a citation by Morfeld et. al. (2016) is not listed in the references.
Response: Thanks for pointing out. We apologize for this mistake in our last manuscript and added the citation by Morfeld et al. (2016) in the text and the References section of the modified version.
- Line 98: I suggest changing from “meals served” to feeding intervals….
Response: Thanks for pointing out. Here we replace “meals served” by “feeding intervals” in our latest submission.
- Line 124: I suggest adding an asterisk in the table to which refers to the current asterisk.
Response: Thanks for pointing out. Asterisks in tables are usually used to indicate the significance level of statistical test results, but the content after the asterisk below Table 1 is just a supplementary explanation. Here we have modified Table 1 in our latest submission.
- Who obtained the Photographs and videos? ‘We’ was used but later only the first author provided a BCS for the manuscript. If the person or persons who obtained the photographs was the first author, it would be an improvement to the credibility to have more than one person score the elephants and combine scores. In addition, why were videos obtained? There is no mention of using the videos in the manuscript.
Response: Thanks for your feedback. We apologize for the lack of clarity in our previous submission. Our study used photographs and video screenshots captured by multiple authors from various angles to assess body condition. All images were scored by the first author only, and other authors were responsible for data collection, such as calculating the proportion of high-calorie feed (%), outdoor enclosure area measurement, recording outdoor time and foot disorders, etc. To clarify, we have added an extra explanation for this in our latest manuscript.
- In the same paragraph each elephant was scored twice (once on each side} by the same person. Adding a description of how the photographs were presented for scoring would help in clarifying the method of determining BCS. Were the photographs presented randomly for scoring without identification, and later decoded for analysis. A question remains, were the images of the same elephant sequentially presented?
Response: Thanks for your feedback. We apologize for the lack of clarity in our previous submission. Images were randomly presented in body condition scoring of each elephant. Before rating, every individual has been identified based on morphological traits. We have added an extra explanation for this in our latest manuscript for clarification.
- Lines 312-313: There was no data presented which would support the statement in this sentence.I highly request removing this sentence.
Response: Thanks for pointing. We apologize for this misexpression in our previous submission and we have removed this sentence in our latest manuscript.
- Lines 313-317:The statement regarding passing EEHV from Wild Elephant Valley is misstating the information found in the citation of Yang et. al. 2022. EEHV has been found in all Asian elephant populations studied in range countries and was found in the small sample size of wild Asian elephants in China reported by Yang et. al. I highly request removing the sentence from 313-317.
Response: Thanks for pointing. We apologize for this miscitation in our previous submission and we have removed this in our latest manuscript.
By making these changes, I believe this is a manuscript which will adding to the information available on captive Asian elephants in China.
Reviewer 2 Report
Comments and Suggestions for Authors
The value of this project is that in enhancing animal welfare. This is, for me, a quite valuable pursuit.
Overall, I would like to see elements of this paper "beefed up". Please consider my suggestions:
Notes on: Obesity Prevalence and Associated Factors in Captive Asian 2 Elephants (Elephas maximus) in China
Line 31: There is the number 6 embedded in the word “countries” i.e. (copy/paste) … “Asian elephants inhabit 13 countr6ies across South and Southeast Asia…..”
Line 43: It is noted the high rate of obesity in captive elephants. Do we know what the incidence/prevalence of obesity is in elephants in the wild?
Line 59: I believe that “dates” should be singular in this sentence; i.e. “China’s history of raising Asian elephants in zoos dates back to 1907” due to the plural of elephants and zoos.
Line 88: The word “in” is missing i.e. “Classification of age groups was referred to the criteria in Arivazhagan and Sukumar (2008). It should read “Classification of age groups was referred to in the criteria in Arivazhagan and Sukumar (2008).
Line 134: Was there a second researcher scoring the elephants BCS? I am concerned with reliability (inter-rater reliability)….. i.e. “All images were scored by the first author.”
Line 144: What is this hypothesis based on? “….we hypothesized that larger outdoor enclosures and extended outdoor time, compared to confined indoor spaces, would offer elephants greater room for movement and the expression of natural behaviors….”. That is, having more room doesn’t necessarily translate to the elephants using that space by increasing activity. Couldn’t they have more room and still “just stand around”?
Line 313: It is stated that, “Furthermore, as the captive elephants in Wild Elephant Valley are not entirely isolated from local wild elephant populations, pathogens (e.g., EEHVs) carried by captive elephants could spread into the wild population when roaming in natural forest….” Could not the opposite occur as well? That is, pathogens from the wild population spread into the captive population? And if not, why not?
Line 330: It is stated that, “We recommend that facilities housing elephants implement measures to increase outdoor time.” Should elephants be encouraged to move about, i.e. increase their activity? That is, I may sit on my couch and binge watch Netflix….and I may als be encouraged to go outside. But if I go outside and sit on my porch reading a book, while I may be outside, I am not any more active…. I do appreciate that there was an attempt to reduce stress on the elephants by refraining from direct contact, but were considerations such as recording elephant activity considered?
I would like to see more emphasis on "why" obesity "matters" to elephants. That is, what, other than lowered fertility rates, does it "mean" to the elephant, to the zoo, and to the community at large when an elephant is obese? For example, are there increased health care costs to the zoo? Is the elephant's mortality correlated to obesity? What benefits are there to the zoo, outside of the elephant's reduce risk of obesity, by adopting your suggestions? e.g. will the food bill go down? vet bills decrease?
Author Response
Comments from Reviewer 2:
The value of this project is that in enhancing animal welfare. This is, for me, a quite valuable pursuit.
Overall, I would like to see elements of this paper "beefed up". Please consider my suggestions:
Notes on: Obesity Prevalence and Associated Factors in Captive Asian Elephants (Elephas maximus) in China
- Line 31: There is the number 6 embedded in the word “countries” i.e. (copy/paste) … “Asian elephants inhabit 13 countr6ies across South and Southeast Asia…..”
Response: Thanks for pointing out. We have fixed this typo in our latest submission.
- Line 43: It is noted the high rate of obesity in captive elephants. Do we know what the incidence/prevalence of obesity is in elephants in the wild?
Response: Thanks for your comments. Due to differences in multiple dimensions, such as movement amount, food availability, and even pressures from predators or poachers, we rarely compare obesity or BCS between wild and captive elephants. There is currently a lack of large-scale studies on obesity status in wild elephants. However, in my opinion, obesity is probably not a problem for wild elephants since it is quite challenging for them to survive under starvation in seasons of food shortage.
- Line 59: I believe that “dates” should be singular in this sentence; i.e. “China’s history of raising Asian elephants in zoos dates back to 1907” due to the plural of elephants and zoos.
Response: Thanks for pointing out. We have fixed this grammar mistake in our latest submission.
- Line 88: The word “in” is missing i.e. “Classification of age groups was referred to the criteria in Arivazhagan and Sukumar (2008). It should read “Classification of age groups was referred to in the criteria in Arivazhagan and Sukumar (2008).
Response: Thanks for pointing out. We have fixed this grammar mistake in our latest submission.
- Line 134: Was there a second researcher scoring the elephants BCS? I am concerned with reliability (inter-rater reliability)….. i.e. “All images were scored by the first author.”
Response: Thanks for your feedback. Our study used photographs and video screenshots captured by multiple authors from various angles to assess body condition. All images were scored by the first author only, since other authors were responsible for data collection, such as calculating the proportion of high-calorie feed (%), outdoor enclosure area measurement, recording outdoor time and foot disorders, etc. We apologize for the lack of multiple raters and consideration of inter-rater reliability in the methodology, and we would pay more attention to this in our future research.
- Line 144: What is this hypothesis based on? “….we hypothesized that larger outdoor enclosures and extended outdoor time, compared to confined indoor spaces, would offer elephants greater room for movement and the expression of natural behaviors….”. That is, having more room doesn’t necessarily translate to the elephants using that space by increasing activity. Couldn’t they have more room and still “just stand around”?
Response: Thanks for your comment. As we mentioned in the third paragraph of "Enclosures, Exhibitions, and Obesity of Captive Elephants in Zoos", based on Ree's study in 2009, zoo elephants spent only 6-19% of 6.5 hours of daily outdoor time walking. However, even so, we still consider that compared with small and boring indoor spaces, larger outdoor enclosures provide elephants with more opportunities to present their natural behaviors, including walking. From the perspective of animal welfare, we also encourage zoo managers to provide larger enclosures for their elephants.
- Line 313: It is stated that, “Furthermore, as the captive elephants in Wild Elephant Valley are not entirely isolated from local wild elephant populations, pathogens (e.g., EEHVs) carried by captive elephants could spread into the wild population when roaming in natural forest….” Could not the opposite occur as well? That is, pathogens from the wild population spread into the captive population? And if not, why not?√
Response: Thanks for your comment. Another reviewer also pointed it out. Studies show that EEHV has been found in Asian elephant populations studied in almost all 13 distribution countries, and it has co-evolved with the Asian elephant. In other words, EEHV is supposed to be omnipresent in free-ranging populations. We apologize for the miscitation of Yang et al. (2022) here, and we have removed these sentences in our latest manuscript.
- Line 330: It is stated that, “We recommend that facilities housing elephants implement measures to increase outdoor time.” Should elephants be encouraged to move about, i.e. increase their activity? That is, I may sit on my couch and binge watch Netflix….and I maybe encouraged to go outside. But if I go outside and sit on my porch reading a book, while I may be outside, I am not any more active…. I do appreciate that there was an attempt to reduce stress on the elephants by refraining from direct contact, but were considerations such as recording elephant activity considered?
Response: Thanks for your comment. Unfortunately, it is often difficult to control the behavior of captive wildlife, especially elephants, the largest land animals today. Even worse, it shows that captive elephants are less likely to exercise or move at all when they become overweight. Given the current situation in most Chinese zoos, the only strong recommendation we can make is to urge keepers to supervise overweight elephants to complete at least one hour of exercise every day if weather permitting, which has been added to our latest manuscript.
I would like to see more emphasis on "why" obesity "matters" to elephants. That is, what, other than lowered fertility rates, does it "mean" to the elephant, to the zoo, and to the community at large when an elephant is obese? For example, are there increased health care costs to the zoo? Is the elephant's mortality correlated to obesity? What benefits are there to the zoo, outside of the elephant's reduce risk of obesity, by adopting your suggestions? e.g. will the food bill go down? vet bills decrease?
Response:
Thanks for pointing out. In latest manuscript, we have added that how obesity can harm the health of captive elephants, such as inducing arthritis and even disability, as well as raising up the risk of of dystocia and stillbirth for pregnant elephants in captivity.
Reviewer 3 Report
Comments and Suggestions for Authors
1. The study addresses an important issue in captive elephant management, focusing on obesity prevalence and associated factors in Asian elephants in China.
2. The use of body condition scoring (BCS) as a method for assessing obesity status is appropriate and well-established in the field.
3. The research identifies key factors influencing obesity, including age, diet, and exercise, which are crucial for developing effective management strategies.
4. The study's findings have significant implications for elephant welfare and conservation, potentially informing improved captive management practices.
5. The inclusion of foot disorders observation adds valuable insight into potential health issues related to obesity and management practices.
Recommendations for Improvement:
1. Consider expanding on the specific dietary and exercise interventions that could be implemented based on the study's findings to combat obesity in captive elephants.
2. Provide more detailed information on the methodology used for body condition scoring, including any potential limitations or biases in the assessment process.
3. Include a comparative analysis with obesity rates and management practices in other countries or regions to provide a broader context for the findings.
4. Explore the potential long-term health impacts of obesity on captive Asian elephants, drawing from existing literature or suggesting areas for future research.
5. Discuss the feasibility of implementing the recommended changes in captive elephant management within the context of Chinese zoos and wildlife facilities, considering potential challenges and resource requirements.
Author Response
Comments from Reviewer 3:
- The study addresses an important issue in captive elephant management, focusing on obesity prevalence and associated factors in Asian elephants in China.
- The use of body condition scoring (BCS) as a method for assessing obesity status is appropriate and well-established in the field.
- The research identifies key factors influencing obesity, including age, diet, and exercise, which are crucial for developing effective management strategies.
- The study's findings have significant implications for elephant welfare and conservation, potentially informing improved captive management practices.
- The inclusion of foot disorders observation adds valuable insight into potential health issues related to obesity and management practices.
Recommendations for Improvement:
- Consider expanding on the specific dietary and exercise interventions that could be implemented based on the study's findings to combat obesity in captive elephants.
Response: Thanks for your feedback. Here we add suggestions based on dietary and exercise interventions to release obesity and overweight of captive Asian elephants in China.
- Provide more detailed information on the methodology used for body condition scoring, including any potential limitations or biases in the assessment process.
Response: Thanks for your comment. We apologize for the lack of description on potential related factors of BCS in our last submission. We have fixed it and also added detailed explanation of body condition scoring in our latest manuscript for clarification.
- Include a comparative analysis with obesity rates and management practices in other countries or regions to provide a broader context for the findings.
Response: Thanks for your precious suggestion. We hope that this work can serve as a reference for studying the obesity status of captive Asian elephants in China, and we will look up international experience to pursue further research in the future.
- Explore the potential long-term health impacts of obesity on captive Asian elephants, drawing from existing literature or suggesting areas for future research.
Response:
Thanks for pointing out. In latest manuscript, we have added that how obesity can harm the health of captive elephants, such as inducing arthritis and even disability, as well as raising up the risk of of dystocia and stillbirth for pregnant elephants in captivity.
- Discuss the feasibility of implementing the recommended changes in captive elephant management within the context of Chinese zoos and wildlife facilities, considering potential challenges and resource requirements.
Response:
Thanks for your advice. In our latest manuscript, we propose feasible suggestions in the “Conclusion” to help alleviate obesity in captive Asian elephants based on the current situation in most Chinese zoos.
Round 2
Reviewer 1 Report
Comments and Suggestions for Authors
Thank you for addressing the concerns expressed regarding the original manuscript. In adding a citation, on line 60 you have incorrectly attributed the authorship of the citation. It should be corrected from (Dennis, S. 2008) to Schmitt, Dennis or Schmitt D. L. In addition the reference should be corrected as well. The rest of the citation is correct.
Author Response
Comment from Review 1 (Round 2):
Thank you for addressing the concerns expressed regarding the original manuscript. In adding a citation, on line 60 you have incorrectly attributed the authorship of the citation.
It should be corrected from (Dennis, S. 2008) to Schmitt, Dennis or Schmitt D. L.
In addition, the reference should be corrected as well. The rest of the citation is correct.
Response:
Thanks for pointing out! We apologize for this template mistake of citation in our former submission and have fixed it in our latest manuscript.
Reviewer 2 Report
Comments and Suggestions for Authors
I appreciate your attention to the suggestions and corrections I offered. I also appreciate the manner in which your revised manuscript was presented as it allowed for faster comparison of those corrections.
Author Response
Comment from Reviewer 2 (Round 2):
I appreciate your attention to the suggestions and corrections I offered. I also appreciate the manner in which your revised manuscript was presented as it allowed for faster comparison of those corrections.
Response:
Thanks for your valuable suggestions on our research. We will seriously consider them and apply these insightful viewpoints to future studies.